# Modern In Vitro Techniques for Modeling Hearing Loss

**DOI:** 10.3390/bioengineering11050425

**Published:** 2024-04-26

**Authors:** Jamie J. Shah, Couger A. Jimenez-Jaramillo, Zane R. Lybrand, Tony T. Yuan, Isaac D. Erbele

**Affiliations:** 1Department of Pathology, San Antonio Uniformed Services Health Education Consortium, JBSA, Fort Sam Houston, TX 78234, USA; couger.a.jimenezjaramillo.mil@health.mil; 2Division of Biology, Texas Woman’s University, Denton, TX 76204, USA; zlybrand@twu.edu; 3Uniformed Services University of the Health Sciences, Bethesda, MD 20814, USA; tony.yuan@usuhs.edu (T.T.Y.); isaac.d.erbele.mil@health.mil (I.D.E.); 4Department of Otolaryngology, San Antonio Uniformed Services Health Education Consortium, JBSA, Fort Sam Houston, TX 78234, USA

**Keywords:** otic organoids, sensorineural hearing loss, stem cells, operational medicine, inner ear modeling, cochlear hair cells

## Abstract

Sensorineural hearing loss (SNHL) is a prevalent and growing global health concern, especially within operational medicine, with limited therapeutic options available. This review article explores the emerging field of in vitro otic organoids as a promising platform for modeling hearing loss and developing novel therapeutic strategies. SNHL primarily results from the irreversible loss or dysfunction of cochlear mechanosensory hair cells (HCs) and spiral ganglion neurons (SGNs), emphasizing the need for innovative solutions. Current interventions offer symptomatic relief but do not address the root causes. Otic organoids, three-dimensional multicellular constructs that mimic the inner ear’s architecture, have shown immense potential in several critical areas. They enable the testing of gene therapies, drug discovery for sensory cell regeneration, and the study of inner ear development and pathology. Unlike traditional animal models, otic organoids closely replicate human inner ear pathophysiology, making them invaluable for translational research. This review discusses methodological advances in otic organoid generation, emphasizing the use of human pluripotent stem cells (hPSCs) to replicate inner ear development. Cellular and molecular characterization efforts have identified key markers and pathways essential for otic organoid development, shedding light on their potential in modeling inner ear disorders. Technological innovations, such as 3D bioprinting and microfluidics, have further enhanced the fidelity of these models. Despite challenges and limitations, including the need for standardized protocols and ethical considerations, otic organoids offer a transformative approach to understanding and treating auditory dysfunctions. As this field matures, it holds the potential to revolutionize the treatment landscape for hearing and balance disorders, moving us closer to personalized medicine for inner ear conditions.

## 1. Introduction

Sensorineural hearing loss (SNHL) is the most prevalent neurosensory deficit globally, affecting approximately 470 million individuals, with an expected increase to 900 million by 2050 [1]. The Veterans Benefits Administration compensation report from 2020 reported that more than 1.3 million veterans were receiving disability compensation for hearing loss [2]. One large study of US military service members found that combat experience was associated with a 63% increased risk for hearing loss [3]. This condition primarily arises from the irreversible loss or dysfunction of the cochlea’s mechanosensory hair cells (HCs) and spiral ganglion neurons (SGNs), which are integral to the auditory process. SNHL, often resulting from traumatic blast injuries, noise exposure, ototoxic medications, genetic predispositions, and infectious diseases, poses a significant challenge to operational medicine. Moreover, the long-term consequences of SNHL, which can affect communication, situational awareness, and overall operational effectiveness, underscore the need for innovative solutions [4,5,6,7,8].

The intricate sensory mechanism of the auditory system hinges on HCs within the cochlea and vestibular apparatus, where they serve as transducers of mechanical stimuli into electrical signals for sound perception and balance, respectively [9,10,11]. The human cochlea starts with approximately 75,000 sensory HCs at birth that diminish over time due to various insults and the natural aging process [5,6,7] (Figure 1). Damage to these cells or the neurons they engage is often irreversible, as mammals possess limited regenerative capacity within the inner ear [12,13,14]. Despite the vast impact of hearing loss and balance dysfunctions that affect more than 6% of the global population, therapeutic measures are predominantly compensatory. Current interventions, such as hearing aids and cochlear implants, only alleviate symptoms without rectifying the underlying causes [15]. This highlights the urgency for innovative approaches to understand and address sensorineural impairments. Moreover, there are no FDA-approved therapies that effectively regenerate lost sensory functions [16,17], predominantly due to the lack of appropriate and effective research models.

The development of in vitro otic organoids as a hearing injury model represents the cutting-edge of research and technological innovation to bridge the therapeutic gap. These three-dimensional multicellular constructs mimic the inner ear’s architecture and functional properties, thereby providing an invaluable platform for research and therapeutic development. Otic organoids have shown potential in several critical areas (Figure 2), including the testing of gene therapies aimed at ameliorating congenital hearing deficits [18,19], which can be crucial for individuals entering operational environments with pre-existing hearing impairments. Secondly, they enable drug discovery initiatives for the regeneration of damaged sensory cells [20], potentially offering therapeutic options for hearing loss resulting from traumatic events or environmental exposures. Lastly, otic organoids serve as invaluable models for understanding the development and pathology of the human inner ear, which can aid in the development of protective measures against noise-induced hearing loss and other environmental factors encountered in operational settings [16,21,22,23]. Their emergence is pivotal, given the stark limitations of animal models in recapitulating the unique human inner ear pathophysiology.

Otic organoids are not only at the frontier of medical research for SNHL but also hold promise for an array of applications, from disease modeling to drug testing, potentially revolutionizing the treatment landscape for auditory and balance disorders. This review article will explore the past and current development and applications of otic organoids in an in vitro setting.

## 2. Methods

A comprehensive literature search was conducted using the PubMed database to identify relevant articles focusing on in vitro techniques utilized in modeling hearing loss. The search was limited to articles written in English. Initial search terms included “in vitro otic organoids”, “sensorineural hearing loss (SNHL)”, “induced pluripotent stem cells (iPSCs)”, “operational medicine”, and “inner ear modeling”. A manual review of the retrieved articles and their reference lists was performed to identify additional relevant publications. Further refinement of the search strategy involved examining articles cited within the retrieved studies. This iterative process ensured a comprehensive collection of pertinent literature for inclusion in the review article.

## 3. Methodological Advances in Otic Organoid Generation

Animal models have provided considerable insights into human inner ear disorders; however, the inaccessibility of the inner ear structure, encased within the temporal bone, along with the limited resolution of non-invasive imaging techniques, impedes direct studies of human hearing pathology [24]. Although mouse models have been a cornerstone in the study of human inner ear disorders, they fall short of replicating the precise developmental timeline of the human cochlea, which matures by the twentieth embryonic week, unlike in mice [7,25]. Moreover, species-specific variations, notably the regenerative abilities observed in non-mammalian vertebrates that are absent in mammals, hinder the extrapolation of animal study results to human cases [26,27]. Consequently, it is essential to establish manipulable and sustainable in vitro models such as otic organoids to enhance our comprehension of the inner ear’s biology and its pathologies [17].

To simulate the human inner ear in vitro, multiple stem cell sources have been harnessed, including human pluripotent stem cells (hPSCs)—encompassing both embryonic stem cells (ESCs) and induced pluripotent stem cells (iPSCs)—as well as tissue-resident adult and fetal progenitor stem cells [28,29,30,31,32,33,34,35,36,37,38,39,40,41,42,43]. These stem cells are pivotal in addressing the scalability challenges associated with patient-derived tissues, which are complicated by the difficulties of biopsy procedures and subsequent cell culture [44,45]. The renewability of hPSCs and their amenability to genetic modifications render them particularly advantageous for introducing or repairing mutations implicated in inner ear diseases [46].

The induction of hPSCs into otic progenitor cells necessitates cell culture protocols that replicate embryonic development stages (Figure 3). These protocols meticulously direct the differentiation of hPSCs to produce the diverse cell types of the inner ear—a multi-lineage organ composed of epithelial, neuronal, and glial cell types derived from the ectoderm, along with the periotic mesenchyme (POM) originating from the mesoderm and neural crest [4,17,46]. This poses a substantial bioengineering challenge, requiring the integration of these distinct cell types into a coherent and functional organoid in vitro [47]. Otic organoids, which are advanced cultures replicating the inner ear’s milieu, provide an innovative platform for the investigation and potential treatment of SNHL. These organoids facilitate intricate analyses of developmental pathways and cellular interactions, which are fundamental in addressing SNHL, and signify a significant advancement over traditional stem cell cultures [17]. Furthermore, otic organoids are instrumental in exploring the human inner ear’s developmental stages and offer insight into hearing loss pathogenesis.

Developing otic organoids has involved both two-dimensional (2D) and three-dimensional (3D) culture techniques (Figure 4). 2D approaches manipulate molecular pathways critical to ear development, resulting in relatively homogenous populations of otic progenitors [32,34,39,48]. While 2D differentiation cultures offer a valuable platform for otic organoid development, they are not without limitations. One prominent constraint is that the generated HCs often exhibit incomplete maturation. These cells may not fully manifest the characteristics and functional attributes of mature in vivo HCs. Moreover, 2D culture systems also face challenges in achieving precise differentiation into diverse HC subtypes, such as cochlear (responsible for auditory perception) and vestibular (responsible for balance) HCs. Lastly, otic organoids cultivated via 2D techniques lack the intricate three-dimensional tissue architecture characteristics of the inner ear [34,39,48].

By contrast, 3D culture methods offer superior mimicry of the complex tissue architecture and cellular interactions characteristic of the inner ear, thereby facilitating enhanced differentiation and maturation of HCs. Researchers have refined protocols that lead hPSCs through a series of sequential developmental stages such as embryoid body formation, ectodermal induction, pre-placodal development, and otic placode formation. These stages are carefully orchestrated to emulate the in vivo development of the ear [49,50,51]. Most notably, a recent study has undertaken a thorough comparison between in vitro inner ear organoids and early in vivo human embryonic otocyst stages, employing advanced methodologies including multiplexed immunostaining and single-cell RNA-sequencing. This comparative analysis has significantly enriched our comprehension of the molecular signatures inherent in these organoids [52]. The manipulation of TGF-β and bone morphogenetic protein (BMP) signaling pathways has been crucial, with fibroblast growth factors (FGFs) and Wnt signaling effectively promoting otic placode development, an essential precursor to the inner ear [37,38,40,50]. Morphological and electrophysiological evaluations, along with gene expression analyses of markers such as Pax2, Myo7a, and Atoh1, have been crucial in determining the similarity of iPSC-derived cells to mature HCs [38,53].

Human iPSC-derived inner ear organoids offer an unmatched opportunity to model the unique aspects of human genetic anomalies related to SNHL. For example, organoids derived from mouse ESCs have shed light on HC degeneration linked to human-specific TMPRSS3 mutations. The CRISPR/Cas9-mediated correction of mutations in genes like Myo7a and Myo15a in iPSCs has demonstrated potential for restoring the structural and functional integrity of stereocilia [54,55,56,57]. The progress with human iPSC-derived inner ear organoids demonstrates the capacity that organoids have to solve complex biological problems by correcting mutations in developed tissue and therefore rescuing the healthy phenotype.

The burgeoning field of 3D inner ear organoids signifies a transformative approach to the treatment of auditory dysfunctions. These structures emulate the cellular organization and functionality of the inner ear, utilizing both tissue-specific stem cells and hPSCs. They are invaluable for elucidating HC types and their interactions with supporting cells, and they pave new pathways for developing therapeutic interventions for SNHL.

## 4. Cellular and Molecular Characterization

Current research in the field of otic organoids involves a comprehensive cellular and molecular characterization to understand their potential in modeling inner ear development and disease pathogenesis. These organoids are generated from hPSCs and recapitulate the complexities of the developing inner ear, providing a valuable platform for studying otic development and disorders. Cellular and molecular characterization efforts have been primarily carried out in both human and animal otic models, shedding light on the markers and mechanisms underlying otic organoid formation and function. Additionally, research has elucidated that multipotent cell populations within the human fetal cochlea and the adult spiral ganglion are exploitable, with iPSCs being particularly valuable due to their ability to differentiate into any cell type and be maintained in culture [38]. iPSCs are pivotal in personalized medicine because of their derivation from patient-specific cells and their immunocompatibility [41,58,59,60]. The differentiation potential of various stem cells, including ESCs and iPSCs, has been recognized, as have their unique capabilities and constraints in differentiating into otic sensory cells [61,62]. Moreover, the inability of the adult mammalian cochlea to spontaneously regenerate HCs poses a significant challenge. In contrast, some regenerative potential has been observed in the neonatal rodent vestibular system and sensory epithelia, a process attributed to pathways such as Wnt and Notch [63,64,65,66,67]. Cells expressing Lgr5 within the sensory epithelia have been identified as promising progenitors for HC regeneration [65].

The development of the inner ear originates from the ectodermal germ layer following gastrulation. During this process, the ectoderm differentiates into neural and non-neural domains, influenced by a BMP concentration gradient [68,69]. Subsequently, this differentiation process results in the emergence of the otic placode within the otic-epibranchial placode domain, which is positioned at the juncture of the non-neural ectoderm and the neuroectoderm. This formation is orchestrated by a plethora of signaling pathways, including those mediated by FGFs, Wnt, TGF-β, BMPs, sonic hedgehog (SHH), and retinoic acid (RA) [70,71,72,73,74,75,76,77,78,79]. Diverse cell lineages, such as epithelial, neuronal, and glial cells, are derived from the ectodermal layer, while the specialized periotic mesenchyme (POM) originates from both the mesodermal layer and the cranial neural crest cells [47].

In both human and animal otic models, extensive cellular and molecular characterization has been performed to identify the key cell populations and molecular markers present in otic organoids. These investigations have revealed the presence of otic progenitor cells marked by the expression of transcription factors such as Pax2, Sox2, and Sox9. These markers are indicative of the early stages of otic placode induction and differentiation, consistent with an intricate developmental process [68,69]. Furthermore, sensory and non-sensory cell populations within human and animal models have been identified through the expression of markers like BRN3C for sensory neurons and GATA3 for non-sensory epithelial cells [43].

At the molecular level, the characterization of these otic organoids entails the identification of biomarkers that signal the transition of stem cells toward an otic fate. Initially, otic progenitor cells are identified using markers such as Pax2 and Pax8, followed by the expression of the transcription factor Atoh1, which heralds the development towards the HC lineage [80]. Additional markers like Myo7A and neurofilament proteins are indicative of sensory cell induction [37]. Throughout organogenesis, a suite of specific gene markers, including Pax2, Pax8, Ecad, Sox2, Lmx1a/b, and Jagged1, delineate the trajectory of prosensory cells [17,79,81]. Neurogenin (Neurog1) and Neurod1 are imperative in the commitment of neuronal progenitors [82], while a composite of gene expressions, encompassing Atoh1, Notch pathway intermediaries (Hes5, Sox2), and Neurod1, orchestrates neurosensory cell development [83,84]. In vitro experimentation has facilitated the generation of otic placodes and otocyst-like structures from iPSCs by modulating the signaling pathways, mirroring the involvement of various signaling pathways such as FGFs, Wnt, TGF-β, BMPs, SHH, and RA during normal otic development [37,70,71,72,73,74,75,76,77,78,79,85]. This represents a significant step forward, proposing a viable method to produce cellular components aimed at SNHL therapies [37]. These organoid systems effectively mimic the differentiation of the otic placode into the otic vesicle, subsequently giving rise to the intricate inner ear structures, which include HCs (inner and outer), supporting cells, and SGNs [37].

Inner and outer HCs are integral components of the cochlea, playing vital roles in auditory transduction. Inner HCs primarily transmit auditory stimuli to the central nervous system, while outer HCs contribute significantly to cochlear amplification, enhancing auditory sensitivity and selectivity. Dysfunction of inner HCs commonly leads to SNHL, while abnormalities in outer HC function can result in cochlear amplification disorders and tinnitus [86]. Modulation of Sonic Hedgehog (SHH) and WNT signaling pathways has been shown to promote ventral gene expression in otic progenitors, mimicking developmental cues for cochlear specification. Moreover, single-cell RNA sequencing has identified key transcriptional pathways, including the involvement of NR2F1, in cochlear differentiation. Structural analyses confirm the development of hair bundles characteristic of inner and outer HCs, with inner HCs exhibiting a U-shaped arrangement. Furthermore, functional assessments demonstrate specific ion conductance typical of mature inner and outer HCs, with outer hair cells expressing membrane-localized PRESTIN [85]. Elucidating the intricate molecular cascades governing the development and maintenance of inner and outer HCs is crucial for identifying potential therapeutic avenues.

While strides have been made in delineating the pivotal transcription factors (e.g., Sox2, Atoh1) and signaling pathways (e.g., Notch) integral to differentiation and regeneration [87,88,89,90], the precise gene expression profile essential for the conversion of iPSCs into HCs remains elusive [91]. Notably, the overexpression of Sox1 may accelerate this process [92]. The regenerative phenomena observed in neonatal rodents offer a beacon of hope for potential treatments in adults; however, replicating the dynamic developmental environment of the inner ear in vitro presents challenges, including variability in the structural and cellular composition of otic organoids [63,65,66,67]. In future research, it would be ideal to identify additional markers that can further validate the model’s accuracy and usefulness. For instance, the identification of markers specific to vestibular HCs, which are responsible for balance and spatial orientation, would enhance the representation of the entire inner ear. Moreover, markers that signify the maturation of various cell types within otic organoids, including synaptic markers for sensory neurons and mechanotransduction-related proteins for HCs, would be valuable for assessing functional maturity.

Challenges and limitations in the cellular and molecular characterization of otic organoids include the need for standardized differentiation protocols to minimize heterogeneity in marker expression among different research groups. Achieving a more accurate recapitulation of the intricate cellular diversity of the inner ear remains a challenge due to the complex interactions and signaling pathways involved in inner ear development. Furthermore, enhancing the functional maturity of otic organoids, especially in terms of achieving functional synapses between sensory neurons and HCs, requires further investigation. Additionally, addressing the ethical concerns and technical challenges associated with obtaining human inner ear tissue for comparison with otic organoids is a hurdle that researchers must navigate.

## 5. Technological Innovations for Culture Enhancement

The field of inner ear organoid research has experienced significant progress, primarily driven by interdisciplinary advancements in three-dimensional (3D) bioprinting and microfluidics technologies. These innovations offer several advantages for the development of inner ear organoids. They excel at creating precise cell configurations, establishing defined signaling gradients, and enhancing cellular arrangement, which closely mirrors the natural ontogenesis of the inner ear [93,94,95,96]. These features contribute to the development of consistent and replicable organoid models.

Technological innovations have significantly advanced the field of inner-ear otic organoid models, primarily through the implementation of 3D cell culture techniques. Protocols for the differentiation of iPSCs into HC-like cells have diversified. Some approaches employ embryoid body formation, followed by adhesion to 2D cultures, while others leverage 3D organoid systems to better mimic in vivo organ architecture [29,31,32,33,39,97,98,99,100]. These protocols typically commence with the inhibition of certain pathways and activation of BMP signaling, progressing to treatments that foster anterior ectoderm development and otic placode induction [31,37,39,97,98,99,101]. The application of extracellular matrix components, such as Matrigel, in 3D cultures has been found to enhance differentiation efficiency [37,38,40,101]. By providing a more physiologically relevant microenvironment, 3D cultures facilitate the recreation of intricate cellular interactions and tissue architectures found within the inner ear. Additionally, 3D cultures allow for the incorporation of multiple cell types, including sensory HCs, supporting cells, and neurons. Validation of the resultant cell types involves the use of molecular markers, morphological assessments, and electrophysiological analyses [29,37,98,99].

Innovations in 3D cell culture techniques have also introduced microfluidics and unique geometries, further enhancing the inner ear otic organoid models. Microfluidics play a critical role by replicating the fluidic milieu of the inner ear, pivotal for the maturation of HCs and the formation of synapses [102]. Microfluidic systems enable precise control over nutrient supply, waste removal, and the establishment of gradients, mimicking the in vivo environment. Enhancements, such as incorporating vascularization within organoids via microfluidic systems, could bolster their semi-physiological relevance and support epithelial polarity over extended culture durations [103,104].

In the inner ear, the blood–labyrinth barrier (BLB) plays an important role in pathophysiology, like the blood–brain barrier (BBB). Incorporating inner ear cell types such as sensory inner and outer HCs, supporting cells, endothelial cells from the stria vascularis, and cochlear neurons into microfluidic systems could facilitate the study of BLB physiology and pathophysiology. Integrating inner ear cells into on-chip platforms can facilitate investigating cellular interactions, transport mechanisms, and barrier function within the context of the BLB [105]. However, several obstacles must be addressed to effectively apply on-chip tools for inner ear research. These include the need for robust differentiation protocols to generate functional inner ear cell types from iPSCs, optimization of culture conditions to mimic the unique microenvironment of the inner ear, and validation of barrier properties in BLB models. Despite these challenges, leveraging on-chip technologies for inner ear research presents exciting opportunities to advance our understanding of BLB physiology and develop targeted therapies for inner ear disorders.

Three-dimensional bioprinting improves the structural fidelity of these models, which is a prerequisite for replicating the cochlea’s tonotopic organization [38,53]. Additionally, the use of unique geometries in culture scaffolds can influence cellular behavior and differentiation, promoting the development of more sophisticated otic organoid models. Progress in this domain has been facilitated by bioengineered scaffolds incorporating extracellular matrix (ECM) components like laminin and collagen. Additionally, the use of decellularized cochlear tissues as scaffolds has shown promise in promoting human cell integration [106,107,108]. However, precise delivery and integration within the cochlea continue to pose significant challenges [29,109,110,111]. The introduction of proangiogenic factors or endothelial cells may lead to the formation of vascularized organoids, which could improve graft viability for in vivo applications [112,113].

The application of CRISPR/CAS-9 technology has been instrumental in enhancing the maturity of inner ear otic organoids and expanding their utility. CRISPR/CAS-9 allows for the precise manipulation of genetic elements, enabling researchers to modify genes associated with otic development and maturation. By optimizing the genetic makeup of organoids, researchers can accelerate the differentiation of cells and improve the overall functionality of these models. Furthermore, CRISPR/CAS-9 has opened doors for studying various genetic disorders affecting the inner ear and evaluating potential therapeutic interventions [54]. Comparative analyses of stem cell therapy and CRISPR/Cas9 gene editing suggest that combined approaches may be efficacious for SNHL treatment [114].

By better recapitulating the native inner ear environment, otic organoid models pave the way for innovative approaches to restoring auditory and vestibular function in individuals with hearing loss or balance impairments. Advancements in pathway manipulation for the proliferation of tissue-specific progenitors have resulted in the generation of “otic spheres” from murine cochlear and vestibular cells, which further differentiate into HCs, or supporting cells [115,116]. Techniques for expansion, such as those developed by McLean et al. (2017), exploit Lgr5-positive supporting cells and modulate Wnt and Notch signaling pathways to facilitate HC differentiation [117]. Moreover, culturing human fetal cochlear progenitors has yielded sensory HCs and supporting cells, furthering the potential of in vitro applications in ototoxicity and regeneration research [43]. These milestones underscore the role of inner ear organoids as a pivotal platform for elucidating and addressing SNHL, setting the stage for precision medicine that aligns with individual genetic predispositions and contributing to drug discovery, disease modeling, and therapeutic innovation [18,118].

## 6. Challenges and Limitations

Within otic organoid research, scientists face a series of technical challenges, confront substantial limitations, and navigate a complex landscape of ethical considerations. One major technical obstacle is the limited efficiency of inducing otic cell fates compared to other ectodermal derivatives, which is essential for the accurate representation of the inner ear’s complex cellular architecture in disease modeling and therapeutic applications [119,120,121]. Furthermore, the adoption of three-dimensional culture systems to facilitate necessary morphogenetic shifts has been difficult, leading to challenges in generating consistent and reproducible organoids [37]. Another significant hurdle is the delivery and functional integration of stem cells into the cochlea for the treatment of SNHL, requiring a deeper understanding of differentiation pathways and critical signaling pathways such as Wnt, β-catenin, and Notch [122,123,124]. Lastly, different cell lines, iPSC vs. ESC, and especially within iPSCs, may have variable responses to induction and maturation.

Despite advancements, limitations persist, particularly in the requirement for sophisticated techniques like 3D bioprinting or microfluidic systems to create precise inner ear organoids [93,94,96,104]. Additionally, while much focus has been placed on neurosensory cell characterization, a comprehensive understanding of all inner ear cell types is still needed to develop more effective treatments and understand ear diseases and injuries thoroughly [119,120,121]. By recapitulating the cellular composition and physiological characteristics of the inner ear, including the stria vascularis, otic organoids hold potential for elucidating the mechanisms underlying age-related hearing loss, such as presbycusis, and other forms of hearing impairment associated with dysfunctions in these cell types. For instance, organoid models have been used to study ion transport mechanisms in the stria vascularis, which are critical for maintaining the endocochlear potential and cochlear homeostasis [38]. Furthermore, the development of co-culture systems involving multiple cell types within otic organoids facilitates the study of cell–cell interactions and the pathogenesis of age-related hearing loss [125]. Ethical considerations are equally pivotal, especially with the use of ESCs, shifting the preference towards hPSCs and iPSCs [123,124]. The direction of research must also consider the origins of stem cells, adherence to research guidelines, and the necessity of animal research.

## 7. Conclusions and Future Direction

In conclusion, this review has highlighted significant progress in the field of human inner ear modeling via hPSCs and iPSCs. Over the past ten years, hPSCs have become pivotal to the advancement of regenerative therapies for the inner ear, even in the face of challenges such as limited culture longevity, variability, and efficiency. The emergence of hPSC-derived otic sensory epithelium models presents extraordinary possibilities for disease modeling and the advent of new therapeutic approaches. Future progress, including the development of vascularized three-dimensional organoids and their amalgamation with microfluidic technologies, is anticipated to significantly refine culture and differentiation techniques, thereby accelerating the preclinical developments for SNHL.

With diligent biological research, single-cell genomics, and sophisticated bioengineering methods, we foresee not only the enhancement of these innovative systems but also their considerable impact on the basic understanding and treatment of inner ear conditions. As this technology matures, the prospects for the application of stem cell therapy and organoid culture in clinical settings become increasingly viable, propelling us toward a future where personalized medicine for inner ear disorders becomes a reality.

## Figures and Tables

**Figure 1 bioengineering-11-00425-f001:**
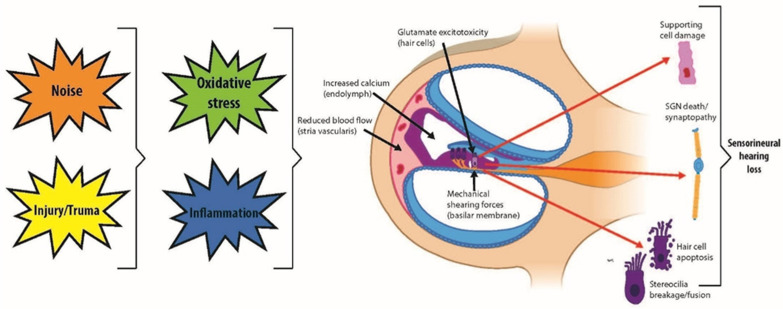
Auditory system demonstrating mechanisms of inner ear hair cell damage and injuries leading to sensorineural hearing loss.

**Figure 2 bioengineering-11-00425-f002:**
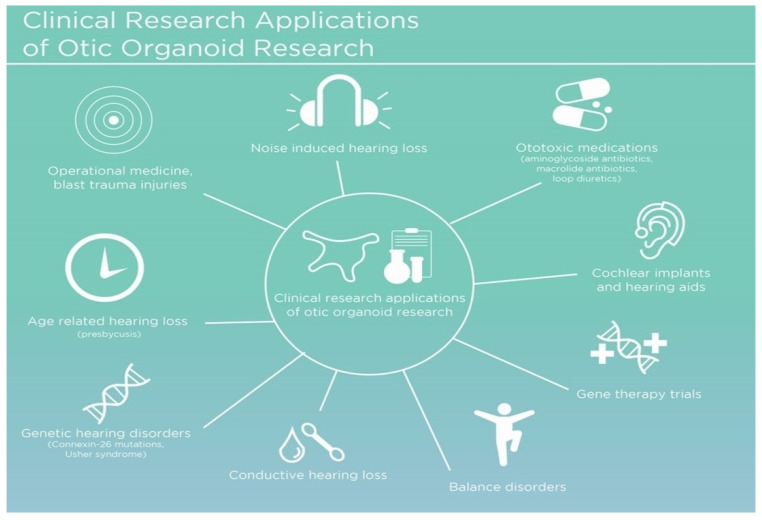
General overview of the various clinical applications for otic organoid research.

**Figure 3 bioengineering-11-00425-f003:**
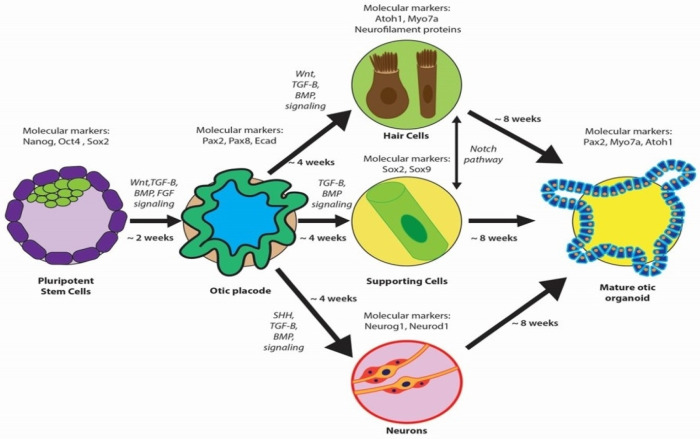
Overview of otic organoid development, highlighting key molecular markers at critical stages. Pluripotent stem cells are induced to form the otic placode and subsequently differentiate into various cell types (hair cells, supporting cells, and neurons), forming a mature otic organoid.

**Figure 4 bioengineering-11-00425-f004:**
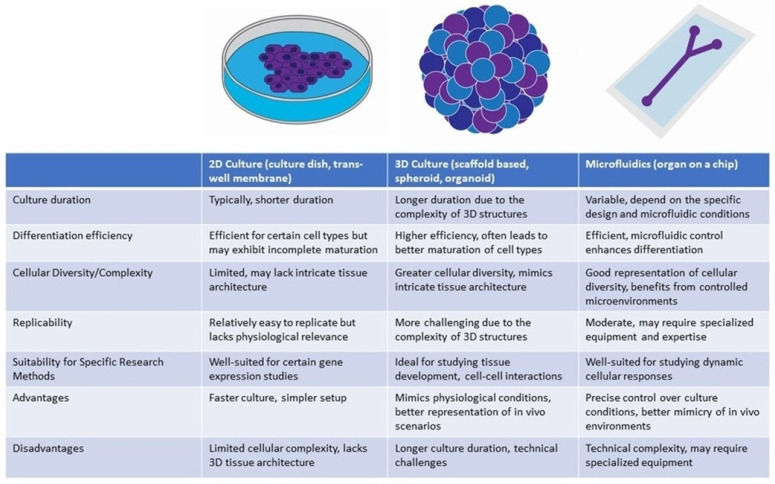
Comparison of otic organoid differentiation methods.

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
