# Peer review of "Modern In Vitro Techniques for Modeling Hearing Loss"

_bioengineering, 2024, doi:10.3390/bioengineering11050425_

Round 1
Reviewer 1 Report
Comments and Suggestions for Authors
The authors describe the literature on inner ear modelling research, with the aim of outlining possible methods to treat inner ear damage (mainly cochlear). However, no frontline of resarch is anywhere near to accomplish this goal, which the authors acknowledge. Yes the authors are hopeful about future results.
The report is both basic in describing the clinical problem with physiology and research methods, as well as going into molecular biology in detail, allowing a top-down perspective of the problem.
What I miss is only a methods section how the literature search was conducted.
Reviewer 2 Report
Comments and Suggestions for Authors
This review addresses an important issue regarding in vitro techniques to create a model for studying hearing loss and identifying new therapeutic targets. The authors focused on in vitro otic organoids as a promising platform for hearing loss models, along with technological innovations to further develop novel therapeutic strategies. This is a well-summarized review and is useful for the advancement of sciences. However, the reviewer would like to suggest a few points for revision to further improve the manuscript.
The authors did not reference recent important papers related to inner ear organoids, such as those published in Cell Stem Cell. 2023;30(7):950-961 and Development. 2023;150(19). Is there a specific reason the authors did not include them?
Regarding lines 130-140 and Figure 4: It is fair and worthy of trust that the authors mentioned about both the advantage and limitation of otic organoids in 2D and 3D culture. However, the reviewer feels that the descriptions are too simplistic. Could the authors provide readers with more information about the specific cell types in the inner ear developed by recent methods? For example, the authors only mentioned “hair cells,” but in the inner ear, there are inner hair cells and outer hair cells with distinct functions. Which cells are developed in otic organoids, and for what purposes (disease types) can they be used? Additionally, there are more than 50 types of cells, including cells in the stria vasularis, that are essential for maintaining homeostasis in the inner ear and are related to hearing loss, including presbycusis. Can the pathophysiology of these cells also be studied using otic organoids generated by recent techniques?
In this field, technological innovation is crucial to create a stable environment. Additionally, since this is the journal “Bioengineering,” it is appropriate and advantageous that the authors addressed technological innervation, including the use of matrigels and microfluids (lines 280-298). Particularly, the reviewer agrees and believes that microfluidic chips are an interesting and promising tool. The references cited by the authors, suc as Cell Stem Cell. 2019;24:995-1005, describe blood-brain barriers with induced pluripotent stem cell (iPSC)-derived brain microvascular endothelial-like cells (iBMECs), astrocytes, and neurons. Since in the inner ear, the blood-labyrinth barrier (BLB) plays important roles instead of the blood-brain barrier (BBB), this human BBB chip could potentially be adapted with modifications for inner ear cells and tissues. Do the authors have any specific ideas for applying these on-chip tools for inner ear research? Which cells in the inner ear are suitable for this application, such as hair cells in the sensory epithelium or endothelial cells in the stria vascularis? Are there any obstacles to the application of inner ear cells? This topic is well-suited for the journal “Bioengineering” and is likely to attract significant interest from readers. Could the authors discuss more details about this topic?
Round 2
Reviewer 2 Report
Comments and Suggestions for Authors
The authors have responded to my comments satisfactory and I think the revised paper now is suitable for publication.